# Male bonobo mating strategies target female fertile windows despite noisy ovulatory signals during sexual swelling

Heungjin Ryu[1,¤a]*, Chie Hashimoto[1,¤b], David A. Hill[2], Keiko Mouri[1,¤b], Keiko Shimizu[3], Takeshi Furuichi[1,¤b]

**1** Primate Research Institute, Kyoto University, Inuyama, Aichi, Japan, **2** Wildlife Research Center, Kyoto University, Kyoto, Kyoto, Japan, **3** Faculty of Science, Okayama University of Science, Okayama, Okayama, Japan

¤a Current address: Graduate School of Informatics, Kyoto University, Kyoto, Kyoto, Japan
¤b Current address: Wildlife Research Center, Kyoto University, Inuyama, Aichi, Japan
* ryu.heungjin.26v@kyoto-u.jp

## Abstract

In most mammals, female sexual receptivity (estrus) closely coincides with ovulation, providing males with precise fertility signals. However, in some anthropoid primates living in multi-male societies, females display extended receptivity along with exaggerated sexual swellings that probabilistically indicate ovulation. This raises the question about how males successfully time mating, particularly when ovulation is difficult to predict from such signals. To address this question in bonobos, we combined daily variation in swelling size, hormonal profiles, and male mating behaviors. By estimating day-specific ovulation probabilities relative to the onset and subsidence (detumescence) of maximal swelling, we also examined how male efforts correlate with female fertility. Our results revealed that while ovulation probability was widely distributed and difficult to predict when aligned with the onset of the swelling phase, male behavior was closely aligned with the conception probability. Males concentrated mating efforts late in the phase and stopped after detumescence. High-ranking males intervened in copulations involving females with higher conception probabilities, specifically those with maximal swelling and older infants. When multiple females exhibited maximal swelling, males preferentially followed females whose maximal swelling started earlier and who had older infants. Male–male aggression increased when there were more females with maximal swelling. However, this tendency was reversed when male party size exceeded the average. Importantly, our results revealed that the low predictability of ovulation is best explained by inter- and intra-individual variation in the length of maximal swelling phase, rather than ovulation occurring randomly within that phase in bonobos. Males effectively manage such a noisy signal by prioritizing late-phase ovulatory cues and integrating reproductive history, thereby extracting usable timing information. This behavioral mechanism helps

**Data availability statement:** All data for statistical models and figures, including R code, are available in the Figshare repository (https://doi.org/10.6084/m9.figshare.30403564).

**Funding:** This study was supported by the Global Environment Research Fund (D-1007 to TF) of the Japanese Ministry of the Environment, the Japan Society for the Promotion of Science (JSPS) Grants-in-Aid for Scientific Research (22255007 to TF; https://kaken.nii.ac.jp/en/grant/KAKENHI-PROJECT-22255007/, and 25304019 to CH; https://kaken.nii.ac.jp/en/grant/KAKENHI-PROJECT-25304019/), and the JSPS Asia-Africa Science Platform Program (2012–2014 to TF). The funders had no role in study design, data collection and analysis, decision to publish, or preparation of the manuscript.

**Competing interests:** The authors have declared that no competing interests exist.

**Abbreviations:** EIA, enzyme immunoassay; GLMM, generalized linear mixed model; LM, linear model; VIF, variance inflation factor.

explain the persistence of conspicuous yet noisy ovulatory signals in bonobos. Since males are capable of inferring ovulation timing even under noisy conditions, selection may not favor highly precise female signals. Instead, it shifts more of the time and energy costs onto males, allowing conspicuous female traits to be maintained.

## Introduction

Darwin's theory of sexual selection [1] explains how traits that appear costly for survival can evolve in one sex if they give advantages in mate competition or attraction [2,3]. Under this scenario, males are typically the sex that more frequently develops conspicuous signals of competitive ability, and females are often more selective about their mates [4–7]. Conspicuous signals, including ornamentation and olfactory chemicals, can impose costs on females by diverting energy from reproduction or by increasing predation risk [8–10]. Particularly in mammals, such costly displays by females are considered rare due to their greater reproductive investment, including long gestation and extended maternal care [11,12]. Female signaling has therefore been viewed mainly as a means of advertising fertility or reproductive condition over short time windows that could influence male–male competition—either by reducing it, affecting its timing, or inciting it—rather than as prolonged displays of competitive ability [5,13,14]. Even so, growing evidence shows that female signaling can provide competitive advantages over other females in mammals [7,14–16]. For example, in cooperative breeding meerkats, females show pronounced aggression that suppresses reproduction in other females, which parallels male–male competition [17]. Nevertheless, even in these cases, females typically compete for reproductive opportunities or for raising offspring, rather than for direct access to mates [15,17], suggesting that a major role of female conspicuous signaling can be explained from the perspective of indirect female choice [18].

Primates differ from most mammals in their heavy use of vision for daily activities and communication [19–21]. In contrast, many other mammals rely primarily on olfactory (chemical) signals [14,22], a pattern linked to nocturnal activity in much of the clade [23,24]. Primates also engage in sexual activity beyond the narrow fertile window [25], whereas sexual behavior in most mammals is largely confined to the narrow fertile windows (estrus) [26]. Among primates, anthropoids (New World and Old World monkeys and apes), exhibit hyper sexual activity independent of ovulation and possess trichromatic color vision that likely evolved for foraging efficiency [27,28]. This visual capacity has then supported the evolution of visual sexual signals, with both sexes exhibiting visually conspicuous secondary sexual traits [29–33].

One of the most distinctive visual signals expressed by female primates is the exaggerated sexual swelling—conspicuous changes in the size and color of the ano-genital skin linked to the ovulatory cycle—found in Old World primates [29,34,35]. It occurs mainly in multi-male and multi-female groups in which females mate with multiple males within a menstrual cycle and appears to have evolved at least three times independently [34,35]. In contrast to what is found in most mammals, it is females,

not males, that exhibit such conspicuous and physiologically costly signals. Several hypotheses have been proposed to explain its role in sexual selection. The best-known hypotheses are the female quality indicator hypothesis (often called reliable indicator [36]) and the graded ovulatory signal hypothesis (also called graded-signal [29]). The female quality indicator hypothesis proposes that variation in swelling size between females reflects differences in reproductive quality, leading males to preferentially invest in females with larger swellings, and is therefore assumed to be shaped by female–female competition for access to males [36,37]. This contrasts with many other mammals, where females more often compete for breeding opportunities or ecological resources rather than mates [17]. Although one study of wild olive baboons reported that larger swellings predicted higher female reproductive quality [37], the analysis was later criticized [38], and subsequent work in the same species did not replicate the result [39]. The graded ovulatory signal hypothesis instead proposes that swelling provides a probabilistic visual signal of ovulation to males, shaping male mating effort and competition within multi-male society [29–31]. A substantial body of research supports key predictions of the graded ovulatory signal hypothesis, showing evidence that swelling size increases with the probability of ovulation, and male mating effort and competition are concentrated at the peak of swelling size when ovulation most likely coincides [40–44]. For example, in mandrills and crested macaques, individual differences in swelling size do not indicate female reproductive quality, but signal their ovulation probability and influence male mating interests [45,46]. In Barbary macaques, changes in swelling size also indicate ovulation probability and increase male copulatory behavior [47,48].

Despite this progress, further investigation is necessary to better understand the relative importance of mate choice and male coercion in determining fertilization, and how the conspicuity and reliability of female signals influence mating and social behaviors. Selection favoring greater signal conspicuity likely carries physiological costs that could otherwise be allocated to reproduction, and weaker coupling between the signal and the timing of ovulation increases the risk of mating-decision errors even for females [25,31]. There is also marked variation among species in how exactly swelling predicts ovulation based on swelling progress. For example, in chimpanzees, swelling size is closely aligned with ovulation timing and elicits male copulation attempts, in line with the graded ovulatory signal hypothesis [49]. In contrast, in bonobos, ovulation is poorly predicted by swelling onset, which may make the signal among the least reliable in species that exhibit exaggerated swellings [50,51]. The causes and consequences of this species variation are unresolved. They may reflect species-specific female adaptations that influence male mating strategies [25,52], measurement artifacts that could inflate noise [51], or broader social functions, including roles in female–female interactions as observed in bonobos [53,54].

In this study, we tested the signal reliability of sexual swelling as a graded ovulatory signal in a wild bonobo group. Bonobos occupy a particularly informative position for the study of mammalian sexual signaling. Unlike humans and most apes, they retain conspicuous ovulatory signals, and females hold high social status and exert substantial control over mate choices and male–male competition, which contrasts with chimpanzees and humans [52,55]. Bonobos also have unusually long maximal swelling phases, resume swelling cycles relatively early postpartum, and therefore experience many cycles before conception [50,56–59]. Estimates suggest more than 20 swelling cycles per conception, roughly twice that in most chimpanzee populations [30,60], with the exception of Taï chimpanzees (11.7~19.4 cycles) [32]. The estrus-sex-ratio hypothesis predicts that frequent swelling appearance lengthens receptive periods and increases the number of receptive (estrous) females per male, thereby reducing male–male competition and increasing mating opportunities, including for low-ranking males in bonobos [52,57]. Additionally, since ovulation can occur outside the maximal swelling phase in bonobos [50,58], but also see [59,61], ovulation is assumed to be difficult to predict from swelling onset, making it a noisy signal [50,51]. This view aligns with the notion that prolonged and imprecise signaling can promote paternity confusion and ease male–male competition [52] by making any single cycle less attractive to males [25,30,62]. However, field data showing substantial male–male competition over mates [55] and strongly skewed paternity toward highest-ranking males [63,64], contradict the low predictability of ovulation from the signal in bonobos. It also raises the question of why females maintain such a potentially costly and prolonged ovulatory signal that may hinder females' ability to

choose high-ranking males, given that prolonged receptive periods can increase sperm competition [65]. Finally, if ovulatory signals are very unreliable due to excessive noise, the signaling system should be unstable and possibly collapse, unless the recipient of the signal is manipulated or counterstrategies have not yet evolved [66–68].

To examine the signal reliability of sexual swellings as a graded ovulatory signal, we first investigated whether male mating efforts are correlated with sexual swelling and the probability of ovulation, particularly to the 4-day fertile window—3 days preceding ovulation and the day of ovulation [69]. We did not expect that males could pinpoint the fertile windows within a menstrual cycle. Instead, we expected that male sexual behavior, particularly that of high-ranking males, would coincide with the fertile windows, as predicted by the graded ovulatory signal hypothesis [29]. Second, we calculated day-specific ovulation probability from the onset of the maximal swelling and compared it with day-specific ovulation probability that was calculated from detumescence. We then aligned these probabilities with male mating efforts. Lastly, based on our findings, we discussed whether maximal sexual swelling serves as a graded ovulatory signal influencing male mating efforts, and whether the low predictivity of ovulation, calculated from the equation (see Materials and methods), is biologically meaningful in bonobos or not. By doing so, this study contributes to a better understanding of how seemly noise signals in bonobos have evolved and how they fit into the evolutionary continuum of the exaggerated sexual swelling in primates and signal communications surrounding ovulation in mammals.

## Results

### Males followed females with maximal swelling who had older infants

To test males' ability to estimate ovulation probability, we first investigated the number of males exhibiting intensive following directed at females—defined as following a target female for over 5 min within a 10 m distance. In total, male intensive following of females was observed on 95 out of 250 observation days. The number of males engaging in intensive following on a given day ranged from 0 to 8 (1.4 ± 2.1 males per day). Of these 95 days, there were 6 days when two females were simultaneously the targets of male intensive following. More males exhibited intensive following of females with maximal swelling and an older infant (Fig 1A and 1B; generalized linear mixed model [GLMM]-1A in Table 1). The effect of the number of males and females with maximal swelling and the interaction between sexual swelling and infant age were not significant.

Males focused their intensive following on females whose maximal swelling was nearing detumescence and whose infant was 3 years old or older—a significant interaction between days from detumescence and infant age (Fig 1C; GLMM-1B in Table 1). The other explanatory variables were not significant. When there were more than 2 females with maximal swelling, the probability that males chose a female for intensive following was higher for those with an older infant and whose maximal swelling phase started earlier (Figs 1D and S1; GLMM-1C in Table 1). However, the number of males did not influence male choice, indicating that males do not alter their preference based on the presence of same-sex competitors, nor do they shift attention to the second most attractive female. Although female age was significant in this model, it may be due to the limited variation in female age in the dataset (S1 Fig). We also found, using 14 ovulation-detected menstrual cycles (S2 Fig), that the number of males exhibiting intensive following of females increased when ovulation was imminent (Fig 1E; GLMM-1D in Table 1), and with increasing infant age. The other predictor variables, including the interaction, did not predict the number of males engaged in intensive following.

### The day-specific probability of ovulation and fertility

The day-specific probability of ovulation and fertility were computed using 14 ovulatory cycles from 5 females (S2 Fig). We employed the same equation that used in previous studies of chimpanzees and bonobos [50,69]. The day-specific ovulation probability was broadly distributed from the 8th to 27th days from the onset of maximal swelling phase (Fig 2A). The peak probability was on the 19th day from the onset of maximal swelling phase. This pattern is similar to that of a previous

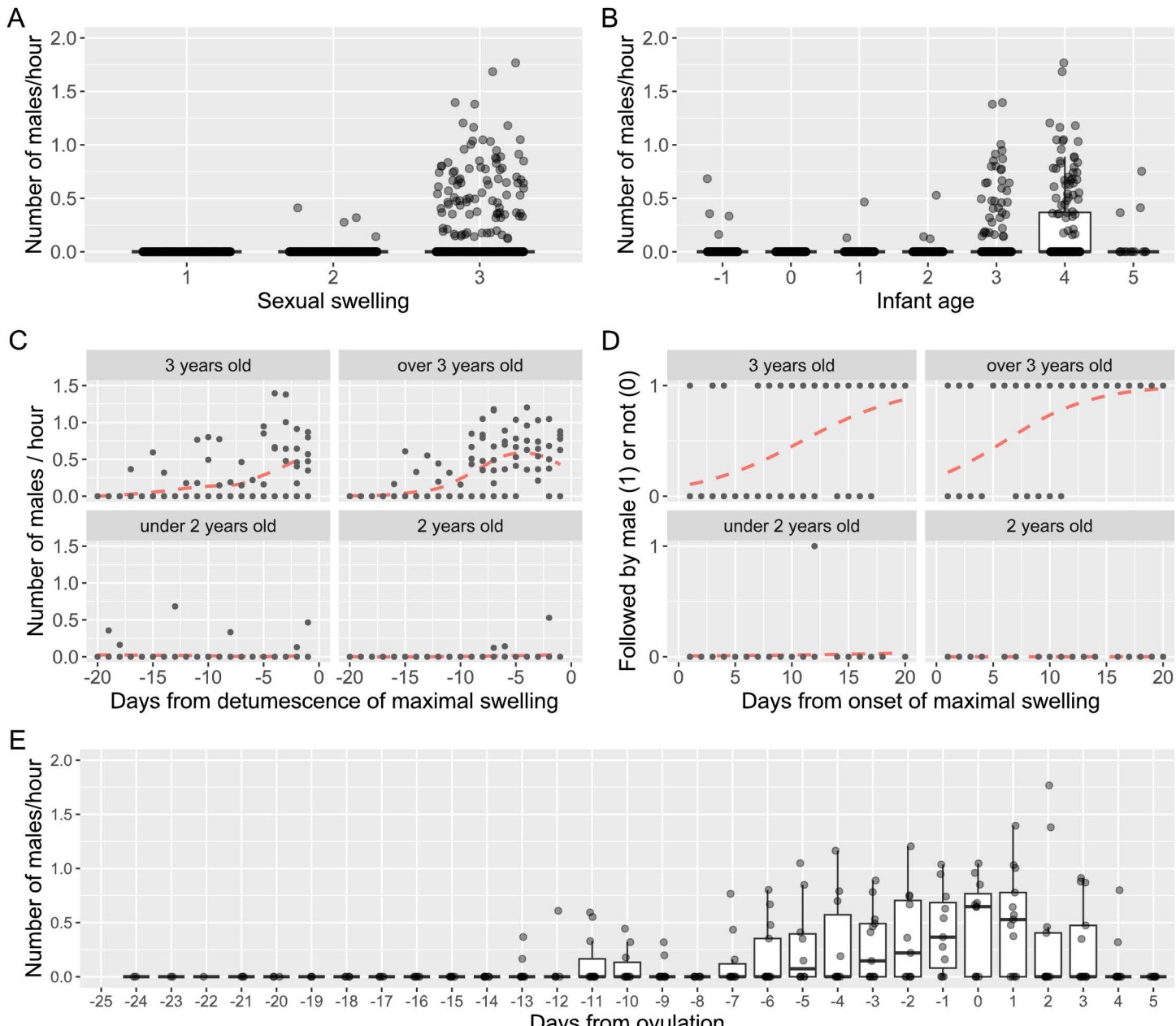

**Fig 1. Males intensive following of females depending on sexual swelling and infant ages. (A)** The number of males exhibiting intensive following increased with sexual swelling size, and **(B)** increased with infant age. Sexual swelling categories were 1 for non-swelling, 2 for intermediate, and 3 for maximal swelling. Infant age ranged from −1 (pregnancy) to 5 years. As the medians were zero due to many zero values, the box plot collapsed to a small box at the 75th percentile with only the upper whisker visible. **(C)** More males exhibited intensive following of females with older infants, especially as the female's maximal swelling neared detumescence. The dashed lines were fitted by "lowess" (locally weighted least squares regression) function. **(D)** The probability that a male performed intensive following increased for females with older infants and with earlier onset of the maximal swelling. The dashed lines were fitted by logistic regressions. Each dot is a daily value. Day 0 is the day of detumescence (C) or one day before the onset of maximal swelling (D). **(E)** The number of males in intensive following increased as ovulation approached. Detumescence started 1.9±2.4 days after the ovulation on average, and the median number of males in intensive following peaked on the ovulation day. The box plot represents the 75th and 25th percentiles with a line indicating the median and whiskers representing 1.5 times the interquartile range. The dots represent all data points. The data and R code underlying this figure can be found in the Figshare repository (https://doi.org/10.6084/m9.figshare.30403564).

**Table 1. Summary statistics of GLMM-1A to 1D.**

| Models | Explanatory variables | Estimates | SE | Z | P |
|---|---|---|---|---|---|
| GLMM-1A (Fig 1A, 1B) | Intercept | −8.31 | 1.07 | −7.77 | <0.001*** |
| | Infant age[a] | 3.09 | 0.75 | 4.12 | <0.001*** |
| | Swelling score[b] | 2.50 | 0.54 | 4.63 | <0.001*** |
| | Female age | −0.16 | 0.70 | −0.23 | 0.819 |
| | Number of males | −0.17 | 0.17 | −1.04 | 0.299 |
| | Number of FMSs | −0.01 | 0.15 | −0.04 | 0.972 |
| | Interaction between [a] and [b] | −0.03 | 0.42 | −0.08 | 0.940 |
| GLMM-1B (Fig 1C) | Intercept | −4.19 | 0.7 | −6.02 | <0.001*** |
| | Infant age[c] | 2.13 | 0.48 | 4.48 | <0.001*** |
| | Days from detumescence[d] | 1.31 | 0.23 | 5.67 | <0.001*** |
| | Female age | −0.26 | 0.7 | −0.37 | 0.714 |
| | Number of males | −0.03 | 0.13 | −0.2 | 0.841 |
| | Number of FMSs | −0.08 | 0.13 | −0.61 | 0.543 |
| | Interaction between [c] and [d] | 1.30 | 0.26 | 4.95 | <0.001*** |
| GLMM-1C (Fig 1D) | Intercept | −2.99 | 0.40 | −7.55 | <0.001*** |
| | Infant age[e] | 3.48 | 0.57 | 6.11 | <0.001*** |
| | Days from onset of MSP[f] | 2.31 | 0.40 | 5.83 | <0.001*** |
| | Female age | 0.85 | 0.40 | 2.14 | 0.033* |
| | Number of males | −0.02 | 0.22 | −0.09 | 0.932 |
| | Interaction between [e] and [f] | 1.91 | 0.47 | 4.11 | <0.001*** |
| GLMM-1D (Fig 1E) | Intercept | −2.97 | 0.38 | −7.87 | <0.001*** |
| | Infant age[g] | 1.59 | 0.42 | 3.83 | <0.001*** |
| | Days from ovulation[h] | 1.74 | 0.23 | 7.59 | <0.001*** |
| | Female age | −0.26 | 0.31 | −0.82 | 0.413 |
| | Number of males | 0.17 | 0.14 | 1.23 | 0.217 |
| | Number of FMSs | 0.10 | 0.13 | 0.78 | 0.434 |
| | Interaction between [g] and [h] | −0.04 | 0.27 | −0.14 | 0.887 |

This table presents the results of GLMM analyses of the number of males who engaged in intensive following of a given female (response variable). All explanatory variables are standardized (mean of 0 and a standard deviation of 1). The maximum variance influence factor (VIF) is 3.44 for the interaction term in GLMM-1A. Letters in superscript indicate the variables also used in interaction terms.

Asterisks indicate significance levels, *: $P<0.05$, **: $P<0.01$, ***: $P<0.001$.

FMSs, females with maximal swelling; GLMM, generalized linear mixed model; MSP, maximal swelling phase.

report on wild bonobos from Lui Kotale [50]. The day-specific probability of fertility—the likelihood that the egg coincides with sperm—did not exceed 0.3 (Fig 2A), which is consistent with the previous report [50].

To investigate the likelihood of ovulation in relation to detumescence and how detumescence impacts male behavior, we also calculated the day-specific probability of ovulation and fertility relative to the day of detumescence [39]. The ovulation probability was concentrated on the day of detumescence and showed a clear peak 4 days before detumescence (Fig 2B), suggesting that progesterone, released from the corpus luteum after ovulation, results in detumescence of maximal swelling [59]. The day-specific fertility exceeded 0.8 from 5 to 4 days before detumescence, then decreased rapidly to zero one day before detumescence. The distribution of the probability of ovulation and fertility, regardless of calculation methods, mirrored the distribution of males exhibiting intensive following on each day. The number of males exhibiting intensive following was spread across days when aligned with the onset of maximal swelling phase (Fig 2C). In contrast, male intensive following was clustered around the days approaching detumescence (Fig 2D). Notably, the

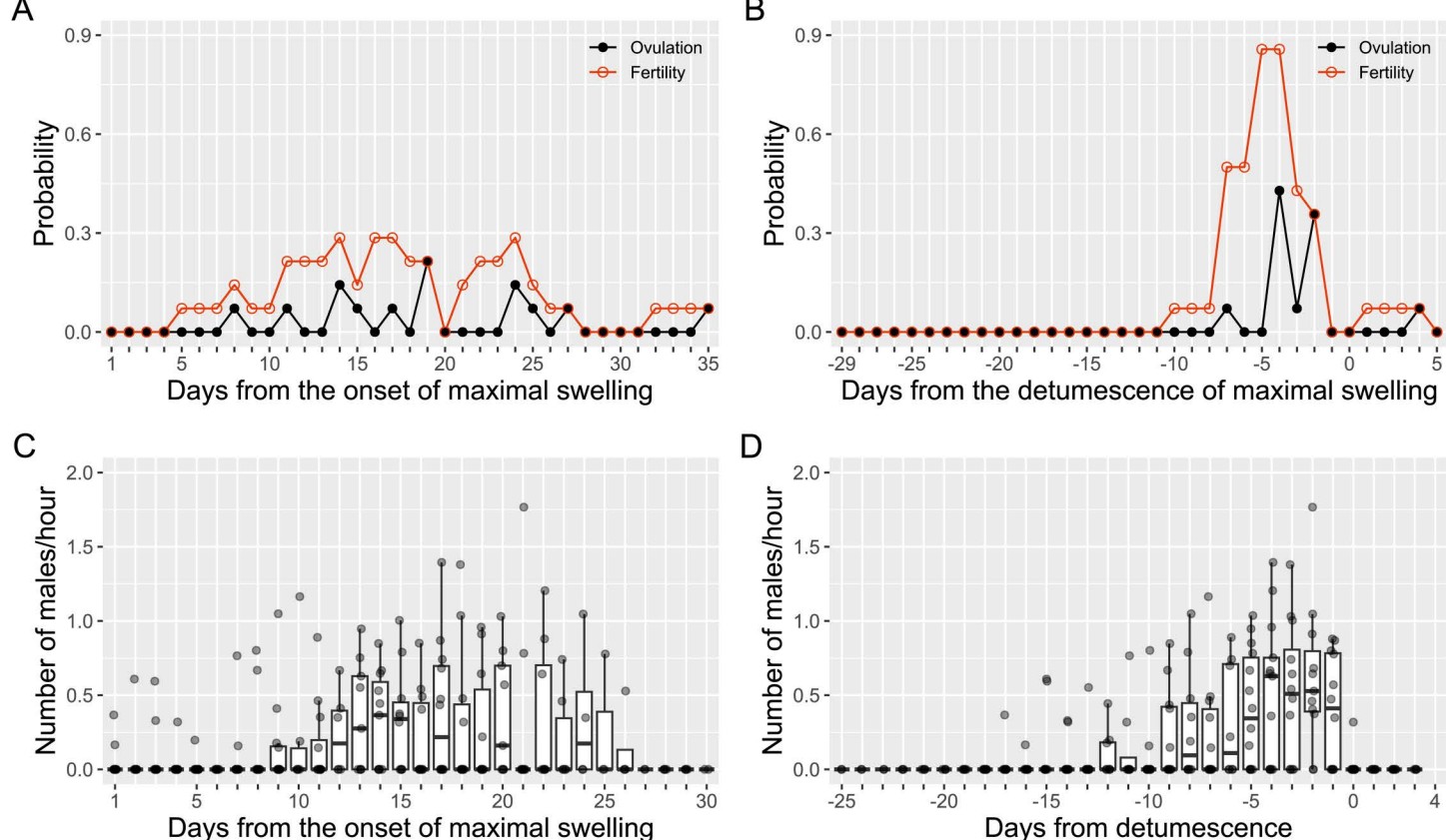

**Fig 2. Probability of ovulation, fertility, and male intensive following. (A)** The day-specific probability of ovulation and fertility from the onset of maximal swelling phase (MSP), and **(B)** from the detumescence of the maximal swelling. Black closed circles and lines represent the day-specific ovulation probability, and red open circles and lines represent the day-specific probability of fertility. **(C)** The number of males exhibiting intensive following of females from the onset of MSP is distributed similarly to the probability of ovulation in A. **(D)** The number of males exhibiting intensive following in relation to the detumescence of maximal swelling shows a similar distribution as found in the probability of ovulation in B. Day 1 is defined as the day of onset of the MSP in A and C. Day 0 is defined as the day of detumescence in B and D. The box plot represents the 75th and 25th percentiles with a line indicating the median and whiskers representing 1.5 times the interquartile range. The dots represent all data points. The data and R code underlying this figure can be found in the Figshare repository (https://doi.org/10.6084/m9.figshare.30403564).

median number of males exhibiting intensive following increased until detumescence, and no males followed females after detumescence, indicating that males use detumescence as a signal to stop intensive following toward the target female. We do not have evidence to suggest that males can predict detumescence and adjust their intensive following.

**Male mating efforts in response to ovulation and male–male aggression**

Males' copulations increased when the ovulation day was imminent (Fig 3A; GLMM-2A in Table 2). Infant age had a negative effect—males copulated more with females with younger infants (Fig 3B), and male rank had a positive effect—high-ranking males copulated more (Fig 3C). Although male age had a negative effect on copulation (Fig 3D), female age had no such effect (Fig 3E). The interaction between infant age and days from ovulation was not significant. To better understand the effect of male rank and infant age on the number of copulations, we further investigated copulatory patterns among males. The top 3 ranking males accounted for 78.9% of all copulations (165/209), and they copulated with

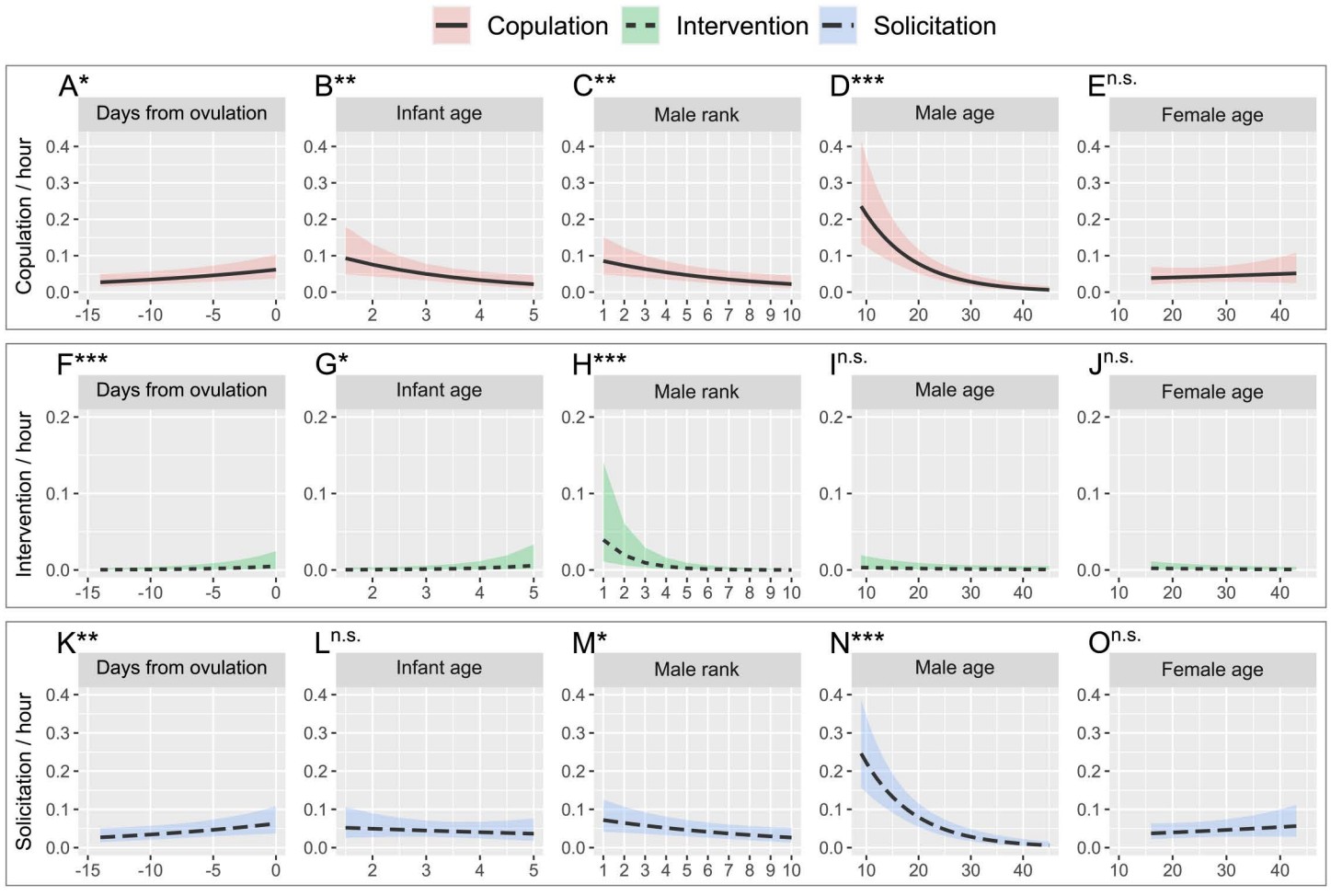

**Fig 3. Prediction plots of males' copulation, intervention, solicitation of copulation in relation to the ovulation day, infant age, male rank, male age, and female age.** (A–E) The predicted copulation rate within a male–female dyad (GLMM-2A). (F–J) The predicted rate of intervention of copulation within a female–male dyad. (K–O) The predicted rate of solicitation of copulation by a male within a male–female dyad. Asterisks (*) on each plot indicate $P$-value (ns: not significant, *: $P < 0.05$, **: $P < 0.01$, ***: $P < 0.001$). The data and R code underlying this figure can be found in the Figshare repository (https://doi.org/10.6084/m9.figshare.30403564).

females with younger infants most often (S3 Fig and GLMM-2B in Table 2), which contrasts with their skewed intensive following of the females with older infants and maximal swelling. Although high-ranking males copulated more frequently with females with young infants (indicating low reproductive quality), this pattern may reflect the reduced mating frequency of females with older infants (indicating high reproductive quality). Due to such skewness in copulation, low-ranking males had fewer chances to copulate even with females of low reproductive quality, which does not support the prediction made by the estrus-sex-ratio hypothesis [52]. The lower copulation rate of females with older infants with high-ranking males indicates that females are more selective when they have higher reproductive quality, and/or males interfere with copulations with each other.

Interventions of copulation between males increased as a female's ovulation day approached (Fig 3F; GLMM-2C in Table 2) and for females with older infants (Fig 3G). High-ranking males intervened more frequently in other males' copulation attempts (Fig 3H). Neither male nor female age predicted the number of interventions (Fig 3I and 3J), suggesting

that the intervention rate was not influenced by female rank, even though older females have a higher rank than younger females.

Males solicited copulation more from females whose ovulation was approaching (Fig 3K; GLMM-2D in Table 2). The effect of infant age was not significant (Fig 3L). Therefore, the higher copulation rate of females with younger infants (Figs 3B and S3) was not due to a higher male solicitation rate directed at those females. Both high-ranking and younger males solicited copulation more frequently (Fig 3M and 3N). Female age was not significant, suggesting that older females are not necessarily more attractive to males, which contrasts with the male preference for older females reported in wild chimpanzees [70].

The hourly rate of male–male aggression (male aggression rate) was higher on days when male intensive following was observed and increased with the number of males present (Fig 4A; LM-2E in Table 2). Although the male aggression rate appeared to increase with the number of females with maximal swelling on the day (Fig 4B), this trend was not significant. However, there was a significant interaction between the number of males and females with maximal swelling on the male aggression rate. To better understand the interaction, we visualized the effect of the number of females with maximal swelling and males on the male aggression rate. As shown in Fig 4C and 4D, the male aggression rate decreased when there were more males and females with maximal swelling in the party. However, when there were fewer males in the party, the effect of the number of females with maximal swelling on the male aggression was reversed, and the male aggression rate increased with an increase in the number of females with maximal swelling. These results demonstrate that the effect of the availability of receptive/attractive females with maximal swelling on the male aggression rates changes depending on the number of males in the party.

## Discussion

In this study, male mating efforts were skewed toward females with higher probability of conception—females with older infants (3 years or older) and those whose maximal swelling was close to detumescence (near ovulation). Males, regardless of rank, concentrated their mating efforts, including copulation interventions, on females with a higher probability of conception. However, high-ranking males accounted for most copulations within the party, indicating priority of access to those females. When there were more than two females with maximal swelling, males followed females with older infants whose maximal swelling phase started earlier, demonstrating males' preference for females with a higher probability of conception. These results suggest males can discriminate conceptive potential of a given female based on her sexual swelling and reproductive history, e.g., infant age. Although male bonobos can discriminate subtle changes in swelling size as found in other primates, including chimpanzees and Barbary macaques [48,49], we could not test this possibility due to our categorical measure. Further investigation using precise measurements of swelling size is necessary to clarify this possibility. However, even in this case, group males who can track daily changes in swelling size until detumescence and retain information about individual variations in swelling characteristics and reproductive history would have greater advantages than outgroup males who cannot utilize such information.

One unexpected finding is that females with younger infants exhibited more frequent copulation, and their mating partners were mostly high-ranking males. This can be explained if male–male competition over females with older infants is intense, resulting in a lower frequency of copulation for those females, or if females with a higher probability of conception copulate more selectively, thereby reducing their copulation frequency. Contrary to a prediction from the estrus-sex-ratio hypothesis, male–male aggression increased when the number of females with maximal swelling increased. However, this effect of the females with maximal swelling on male–male aggression was reversed when the number of males in a party exceeded the mean number of males party (i.e., in very large parties), implying that the effect of the number of receptive females on male–male agonistic interactions is context-dependent and warrants further investigation.

As in a previous study [50], we also found that day-specific ovulation probability was widely distributed across the maximal swelling phase (Fig 2A), suggesting ovulation is difficult to predict from the onset of maximal swelling. However,

**Table 2. Summary statistics of GLMM-2A to LM-2E.**

| Models | Explanatory variables | Estimates | SE | Z | P |
|---|---|---|---|---|---|
| GLMM-2A Copulation (Fig 3A–3E) | Intercept | −4.89 | 0.27 | −18.19 | <0.001*** |
| | Days from ovulation | 0.24 | 0.10 | 2.49 | 0.013* |
| | Infant age | −0.48 | 0.18 | −2.65 | 0.008** |
| | Male rank | −0.44 | 0.17 | −2.65 | 0.008** |
| | Male age | −1.28 | 0.22 | −5.80 | <0.001*** |
| | Female age | 0.02 | 0.19 | 0.11 | 0.911 |
| GLMM-2B Copulation (S3 Fig) | Intercept | −4.07 | 0.30 | −13.61 | <0.001*** |
| | Days from ovulation[a] | 0.30 | 0.10 | 3.01 | 0.003** |
| | Infant age[b] | −0.50 | 0.17 | −2.86 | 0.004** |
| | Male rank (high[c] vs. low[d]) | −1.35 | 0.34 | −3.99 | <0.001*** |
| | Male age | −1.23 | 0.22 | −5.62 | <0.001*** |
| | Interaction between [a], [b], and [c] | −0.42 | 0.14 | −3.01 | 0.003** |
| | Interaction between [a], [b], and [d] | 0.24 | 0.14 | 1.70 | 0.090 |
| GLMM-2C Intervention (Fig 3F–3J) | Intercept | −7.94 | 0.82 | −9.70 | <0.001*** |
| | Days from ovulation | 0.85 | 0.25 | 3.43 | <0.001*** |
| | Infant age | 0.81 | 0.36 | 2.26 | 0.024* |
| | Male rank | −2.14 | 0.57 | −3.77 | <0.001*** |
| | Male age | −0.58 | 0.41 | −1.42 | 0.157 |
| | Female age | −0.47 | 0.25 | −1.93 | 0.053 |
| GLMM-2D Solicitation (Fig 3K–3O) | Intercept | −4.53 | 0.23 | −19.59 | <0.001*** |
| | Days from ovulation | 0.29 | 0.10 | 2.84 | 0.005** |
| | Infant age | −0.10 | 0.15 | −0.68 | 0.499 |
| | Male rank | −0.34 | 0.14 | −2.45 | 0.014* |
| | Male age | −1.25 | 0.19 | −6.43 | <0.001*** |
| | Female age | 0.15 | 0.15 | 1.00 | 0.318 |
| LM-2E Aggression (Fig 4A, 4B) | Intercept | 0.30 | 0.01 | 21.77 | <0.001*** |
| | Intensive following (x: o) | −0.09 | 0.02 | −5.76 | <0.001*** |
| | Number of FMSs[e] | 0.02 | 0.01 | 1.55 | 0.123 |
| | Number of males[f] | 0.05 | 0.01 | 4.69 | <0.001*** |
| | Interaction between [e] and [f] | −0.03 | 0.01 | −3.14 | 0.002** |

This table presents the results of GLMMs (2A to 2D) that used the number of copulations, interventions, and solicitations of copulation between male–female dyads as response variables, collected over a total of 104 days. Days from ovulation include −14 to 0 days from an ovulation. In GLMM-2B, male rank 1st to 3rd coded "high" and 4th to 10th is coded "low". LM-2E shows results of the number of male–male agonistic interactions in relation to the existence of male intensive following of females and the number of females with maximal swelling. All explanatory variables are standardized, and the maximum VIF is 2.07 for the number of males in LM-2E. Letters in superscript indicate the variables also used in interaction terms.

Asterisks indicate significance levels, *: $P < 0.05$, **: $P < 0.01$, ***: $P < 0.001$.

FMSs: females with maximal swelling; GLMM, generalized linear mixed model.

this statistically driven unpredictability may not reflect biological relevance in bonobos. The equation (see Materials and methods and [69]) calculates day-specific ovulation probability by summing probabilities for each day across all defined ovulatory maximal swelling phases in the group. In this case, there are two ways to achieve a wide distribution. One is that ovulation occurs randomly within or outside the maximal swelling phase—truly unpredictable (random) ovulation. The other is that although ovulation occurs near detumescence (Fig 2B)—post hoc trackable ovulation—large inter- and intra-individual variation in the length of the swelling phase (S2 Fig) results in a wide statistical distribution (Fig 2A). As

PLOS Biology

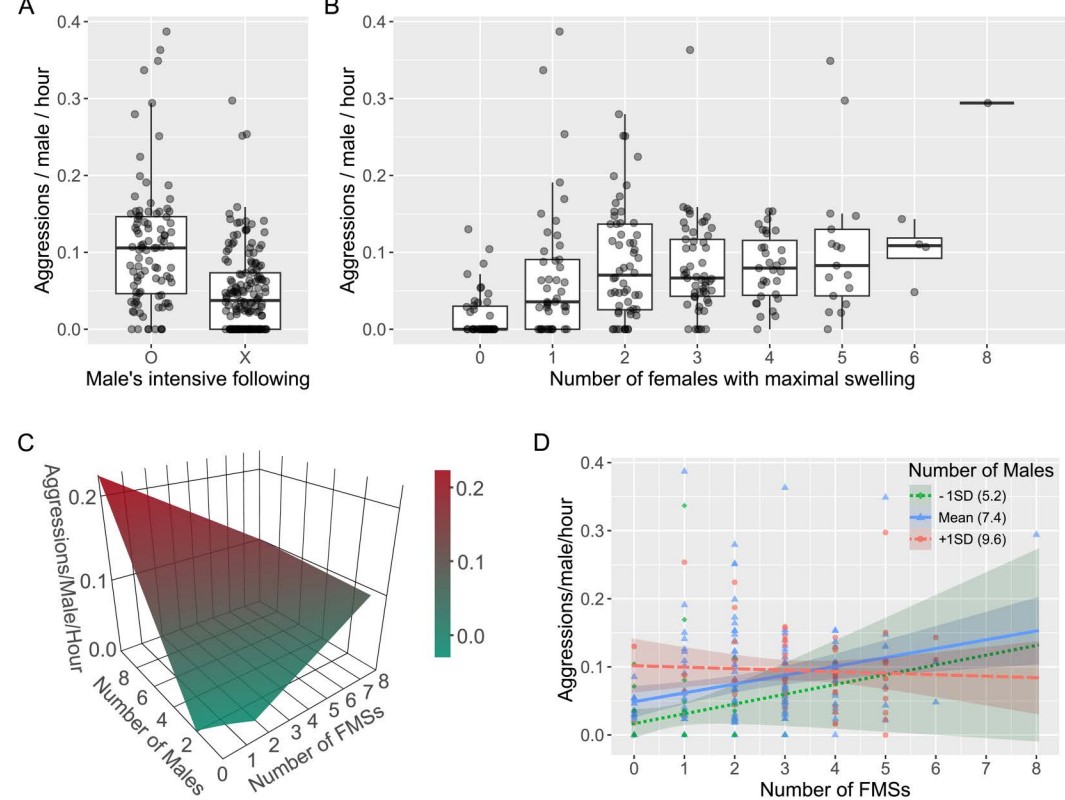

**Fig 4. The number of male–male agonistic interactions in relation to females with maximal swelling. (A)** A higher rate of agonistic interactions per hour between males was observed when males exhibited intensive following. The presence of intensive following is denoted by "O" and its absence by "X". **(B)** The number of male–male agonistic interactions appears to increase with an increase in the number of females with maximal swelling. However, this trend was not significant. **(C)** The effect of the number of males and females with maximal swelling on the hourly male aggression rate based on the model prediction of the LM-2E shows an interdependency between the number of males and females with maximal swelling. This plot was redrawn to fit to the frame based on an interactive 3D plot available at https://doi.org/10.6084/m9.figshare.28416074. **(D)** Visualization of the interdependency between the number of males and females with maximal swelling on the male aggression rate (significant interaction in LM-2E) shows that when there were a greater number of males in the party, e.g., Mean + 1SD (9.6 males in D), the male aggression rate decreased as the number of females with maximal swelling increased. However, when there were a fewer number of males in the party, the male aggression rate increased with an increase in the number of females, with maximal swelling. Each dot represents a single data point, specifically the aggression rate per hour over 250 days. The box plot represents the 75th and 25th percentiles with a line indicating the median and whiskers representing 1.5 times the interquartile range. The data and R code underlying this figure can be found in the Figshare repository (https://doi.org/10.6084/m9.figshare.30403564).

bonobos show longer maximal swelling phases than chimpanzees (Mann–Whitney U Test: $W = 48$, $p = 0.024$; Table 3), the maximal swelling phase likely contains greater variation [25]. Since the maximal swelling phase corresponds to the follicular phase in bonobos [59,61], greater follicular variance can inflate the unpredictability of ovulation in pooled datasets. Physiological factors such as heightened estrogen sensitivity [59] and reproductive events, including lactation, pregnancy, and early pregnancy loss, also increase variability of maximal swelling phases by influencing menstrual cycles [59,71]. Postconception swelling, followed by early miscarriage, as observed in female No (S4 Fig), illustrates how such hormonal fluctuations after implantation can mimic preovulatory signals, adding further variation [72].

Despite the low predictability of ovulation, males concentrate mating efforts on females with a high conceptive probability—females with maximal swelling that started earlier and with older infants—and paternity remains highly skewed toward alpha males [63,64]. Our study suggests that this mismatch between male behaviors and the equation-derived unpredictability of ovulation likely stems from large inter- and intra-individual variations in the length of maximal

**Table 3. The length of maximal swelling phases in female bonobos and chimpanzees.**

| Species | Condition | No. cycles | No. ID | AVG | SD (SE*) | Study sites (reference) |
|---|---|---|---|---|---|---|
| Bonobo | C | 6 | 1 | 15.3 | – | Yerkes [73] |
| Bonobo | C | 9 | 4 | 11.5 | 2.7* | Planckendael [61] |
| Bonobo | C | 23 | 8 | 16.0 | 6.8 | Cologne and 3 sites [58] |
| Bonobo | C | 57 | 4 | 13.4 | 0.7* | Apenheul [74] |
| Bonobo | W | 70 | 13 | 10.6 | 6.8 | Luikotale [50] |
| Bonobo | W | 9 | 3 | 14.6 | 7.4 | Wamba [56] |
| Bonobo | W | 9 | – | 12.9 | 9.3 | Wamba [75] |
| Bonobo | W | 36 | 9 | 14.0 | 11.2 | Wamba [71] |
| Chimpanzee | C | 53 | 13 | 11.9 | 4.4 | Norman, OK [76] |
| Chimpanzee | C | 158 | 10 | 10.4 | – | Yerkes [77] |
| Chimpanzee | W | 41 | 46 | 12.7 | – | Gombe and 2 sites [78] |
| Chimpanzee | W | 33 | 12 | 10.9 | 3.2 | Taï [69] |
| Chimpanzee | W | 27 | 28 | 12.5 | – | Mahale [79] |
| Chimpanzee | W | 37 | 5 | 10.9 | – | Mahale [80] |
| Chimpanzee | W | 37 | 6 | 9.6 | – | Gombe [81] |

The length of maximal swelling phases (MSPs) in captive and wild bonobos and chimpanzees from the available studies. The average length of MSPs from all available studies is $13.5 \pm 1.8$ days for bonobos and $11.3 \pm 1.1$ days for chimpanzees. The length of MSPs of female bonobos is longer than that of female chimpanzees (Mann–Whitney U Test; $W = 48$, $p = 0.024$).

No. cycles: the number of cycles investigated, No. ID: the number of individuals, AVG: average length of MSPs, SD: standard deviation, SE*: standard error, W: wild, C: captivity, –: not available.

swelling phase, rather than from unpredictable ovulation based on swelling progression. As we showed, detumescence provides a reliable postovulatory cue, and males can adjust their mating efforts accordingly (Fig 2D). High-ranking males' reproductive success also demonstrates their mating allocation strategy is effective [63,64,82]. Therefore, our results suggest that male bonobos extract usable ovulation timing from probabilistic, noisy signals by focusing mating effort around detumescence and utilizing reproductive history (e.g., infant age). This male strategy also helps explain how conspicuous but noisy ovulatory signals in bonobos can persist. If males can recover ovulation timing despite noise, selection need not favor highly precise signals. This relaxes the constraints on female signal precision, shifts greater time and energetic costs to males, and still allows conspicuous, graded signal to be maintained.

In most mammals with estrous cycles, sexual behavior is tightly restricted to ovulation, creating high-fidelity signals that reduce male search and guarding costs. By contrast, anthropoid primates often exhibit menstrual cycles with extended receptivity and conspicuous but probabilistic ovulatory signals, including exaggerated sexual swellings. Under multi-male mating, a graded but noisy signal still yields female benefits—reducing male monopolization of mating/paternity and securing social tolerance—without strongly favoring on paternity certainty for high-ranking males, provided that males can utilize integrated information (sexual swelling with postovulatory cues and reproductive events) with sustained mating effort. Thus, our findings specify how a behavioral mechanism can stabilize noisy signaling systems in mammals—male strategies that shift from detecting a brief, reliable estrus to tracking phase-specific landmarks and combining cues over time. More broadly, the equilibrium between conspicuousness and noise of female signals should reflect a balance between (i) the costs and benefits to females of ambiguity, (ii) the energetic and opportunity costs to males of prolonged effort, and (iii) the presence of terminal, postovulatory cues (e.g., detumescence) that signal the end of the fertile window and allow males to disengage. By documenting how bonobo males succeed under noisy signals, we fill the gap between proximate mating behavior and ultimate causation and maintenance of conspicuous, noisy, but probabilistic ovulatory signals across mammals.

The emergence of exaggerated sexual swelling in anthropoid primates extends the duration of ovulatory signaling and facilitates mating across a broader timeline that includes non-fertile periods. However, importantly, even though females exhibit such a conspicuous visual signal, its function is not for female–female competition or ornamentation to indicate their competitive quality. It is rather for attracting male attention and influencing male–male competition, implying the function of such a conspicuous visual fertility signal is not for intrasexual competition (competition over mates) and is still in line with what is suggested by sexual selection in female mammals—female–female competition for reproductive environment and opportunity [4,11,17]. Our findings in bonobos align well with this framework. In this framework, bonobos fall on imprecise end of an evolutionary continuum of conspicuous ovulatory signals, where signals are probabilistic and prolonged. At the other end, there are mammal species with short estrous cycles, where ovulatory signals are conspicuous, but precise, and sexual behavior is tightly confined to the fertile period. By situating bonobo swellings within this evolutionary continuum, we provide insight into how males adaptively allocate mating effort in response to prolonged and probabilistic ovulatory signals, while also noting the importance of postovulatory cues in guiding this allocation. Our study therefore provides evidence that sexual selection has shaped the diversification of female ovulatory signals across mammals, while emphasizing that male mating-effort allocation in response to these signals remains central to the maintenance of such signaling systems.

## Materials and methods

### Ethics statement

The current study was approved by the Ministry of Scientific Research in the Democratic Republic of Congo, under permission numbers MIN.ESURS/SG-RST/13/2013, MIN.ESURS/SG-RST/007/2014, and MIN.ESURS/SG-RST/026/2014. We conducted the study in accordance with the Guidelines for Field Research of Non-Human Primates (Primate Research Institute, Kyoto University, Japan) and the Code of Best Practices for Field Primatology (International Primatological Society).

### Study site and subjects

We conducted this study on a fully habituated, free-ranging wild bonobo group, E1, at the long-term bonobo field site at Wamba (00° 11′ 07.6″ N, 022° 37′ 57.5″ E; WGS84) in the northern sector of the Luo Scientific Reserve, D. R. Congo [83,84]. The group's home range includes primary, secondary, and swamp forests. Despite the absence of a clear division of dry and wet seasons, rainfall and fruit availability are lower in January and February [85]. During the study periods from 2013 to 2015, the E1 group consisted of 32–38 individuals, including 9 adult and 3 adolescent immigrant females, and 7 adult and 4 adolescent males. Age classes were defined following a previous study [86]. We collected focal data on 9 adult females and 7 adult and 3 adolescent males (S1 Table). All adult females had immigrated into the E1 group and had at least one successful birth [87].

### Behavioral observation

We collected behavioral data on adult males and females during three study periods: SP1 (September 2013 to February 2014), SP2 (July 2014 to September 2014), and SP3 (November 2014 to April 2015). We followed bonobos from their bed site, located at around 6 AM, until they made a new night bed, usually after 5 PM. When the group split into several parties, the largest party was followed. To control for the effect of the number of available group members, we recorded party composition every hour using the 1-hour party method [88]. Daily ranging party size and membership were defined as the number and identities of all individuals found during a given day when the total observation time exceeded 2 h.

For continuous focal sampling, we randomly selected one bonobo from the party based on a pre-generated random order before fieldwork began. We searched for the first target focal animal in the random order for at least 30 min after

locating the bonobos in the morning. If we could not find the first one, we chose the second one in the order, and so on. To compensate for the lack of focal sampling data due to the species' fission–fusion social system [89], we prioritized individuals with fewer data for focal animal selection. One focal session continued for 20 min and was terminated if a focal animal was out of sight for more than 5 min within a session. Only focal sessions that lasted longer than 15 min were included in the data analysis. After an individual was followed once, it was not followed again for another focal sampling until at least 100 min had passed. During a focal session, we recorded all activities involving the focal animal, including feeding, moving, resting, and social interactions. We recorded the names and behaviors of individuals within 5 m of the focal animal at 5-minute intervals, including an initial scan at the beginning of each focal session to collect information on nearby individuals. We also recorded all rare events, such as sexual and agonistic interactions, as far as possible. We recorded the individual ID of the participants of the interactions as well as the time and duration. All copulations, genito–genital rubbings, and agonistic interactions (e.g., bite, hit, displacing, and chase) were recorded whenever observed.

Ejaculation during copulation was not always possible to confirm. Therefore, complete heterosexual intercourse without any interruption by others was counted as copulation. Intervention in copulation included direct aggression toward a male who solicited copulation from a female, and direct intervention that targeted the male during copulation. Intervention directed toward females from males was not included, as there were only 22 attempts and most of them led to counter-aggression from females or were otherwise ineffective. Only 5 cases of intervention directed toward females were successful in stopping copulation. Solicitation of copulation was counted only when a clear target was identified. Male bonobos often have an erect penis during feeding while watching females, so only the presence of an erection was not counted as solicitation. For solicitation, an erection should coincide with other behaviors such as directed hand or body gestures [90] or shaking, bending, or dropping branches or twigs to get attention from the target female. We also recorded agonistic interactions whenever there was a clear target, regardless of physical contact. Therefore, undirected display behaviors, such as charging and branch dragging, were not considered agonistic interactions [91]. When a counterattack was performed by the target within 5 s of the directed aggression toward the target, we considered it a tied interaction.

To evaluate males' interests toward females in the party, we defined male intensive following directed at females as when a male maintains proximity (within 10 m maximum) to the target female for more than 5 min while moving, without losing the target female from his sight. If a male's following behavior met these criteria, we considered that this male performed intensive following of the target female.

In total, we followed bonobos for 1462.3 h over 255 days (5.73 ± 1.25 h per day). The total focal sampling time for all individuals was 280.22 h (858 sessions: 14.75 ± 2.01 h per individual). The average length of a focal session was 19.6 ± 1.0 min, and each session consisted of 4.88 ± 0.34 scans (4,189 scans in 858 sessions). We excluded 2 days from the 255 days when we were able to follow bonobos for less than 1 h. Additionally, one day with very heavy rain, when no males were confirmed, and 2 days when we could not see any females, were removed from data analysis, except for the calculation of male dominance relationships.

## Assessment of sexual swelling and the maximal swelling phase

We scored the daily variation in sexual swelling of female bonobos in relation to the firmness and size of each individual's swelling [53,56]. The sexual swelling status was assigned to one of three categories. S5 Fig shows an example of changes in sexual swelling of one individual, named Fk. Swelling 1 is for non-swelling status; swelling 2 is for intermediate swelling; swelling 3 represents maximal swelling. Unlike female chimpanzees, the sexual swelling of female bonobos is always visible, even in the non-swelling status. Therefore, the practical definition of non-swelling status (swelling 1) is that the swelling is within its minimum size range. The swelling size variation of an individual between swelling 2 and 3 was sometimes not definitive. Therefore, firmness and shininess of the surface and movement of the sexual swelling during locomotion were the key features for distinguishing swelling 2 and 3. The detumescence day was set to the first day when

the maximal swelling started shrinking, lost its firmness and shininess, and the swelling score moved from 3 to 2. The maximal swelling phase was defined from the first day of the appearance of the maximal swelling to the end day of the maximal swelling phase before detumescence. If sexual swelling recovers from 2 to 3 within 4 days, it was considered that the maximal swelling phase was continuous, following the previous study [50]. To reduce observation errors and bias in sexual swelling scoring, at least one researcher and two research assistants discussed the sexual swelling status of each female during field observations and decided scores at a daily meeting every evening.

## Urine sample collection and hormonal measurement

Urine samples were collected using filter papers (Whatman #1 Ø 5.5 cm) throughout the day. After a female finished urination, we collected urine from the leaves of terrestrial plants or small trees. We placed the edge of the filter paper (around 1/4 of the whole) into the urine droplets and waited until it absorbed around 2/3 of the paper, allowing us to avoid contamination from dirt on the leaves. To prevent disturbing the bonobos, we collected urine after they had left the area, maintaining a minimum distance of 5 m. We did not collect urine that was mixed with feces or other bonobos' urine. Upon returning to camp, urine-soaked papers were placed in a dry box containing 500 g of silica gel. Although the papers typically dried within 2 days, they were kept in the dry box for a minimum of 5 days. The silica gel was replaced weekly and heated in a hot pan for over 30 min for reuse. Once each paper was removed from the dry box, it was individually packed in plastic zipper bags, then placed in a larger zipper bag with silica gel and stored in a dark room. The longest period that the sample was stored at room temperature was 6 months. After the samples were transported back to Japan, they were stored in a freezer (−20 °C) until the time of urine extraction. A previous report demonstrated that using this method, the estrogen and progesterone metabolite levels do not change for 6–12 months even at room temperature [92,93].

We used an enzyme immunoassay (EIA) to measure urinary metabolites of estrogen ($E_1C$) and progesterone (PdG). In total, we successfully recovered urinary $E_1C$ and PdG from 660 urine samples from 9 females, which were used to determine ovulatory swelling cycles (S1 Table). For the EIA, urine was extracted from the filter paper using deionized water with mechanical shaking [93]. Following extraction, we measured the creatinine concentration of the samples using the Jaffe reaction [94]. When the creatinine concentration of the samples exceeded 3 mg/dl, we retested them by diluting the extracted samples. Urinary $E_1C$ and PdG concentrations were measured by EIA, as described in a previous study [95]. We used antibodies (Cosmo Bio in Japan) against estrone-3-glucuronide BSA, pregnanediol-3-glucuronide BSA, and horseradish peroxidase conjugated steroid derivatives (Cosmo Bio in Japan). More details regarding the EIA are published elsewhere [93,96]. The sensitivity of EIA was 6.6 pg/ml for $E_1C$ and 2.1 ng/ml for PdG. If the concentration was below these values, we excluded that sample from the analysis. The inter-plate CV for $E_1C$ was 10.79 and 15.83 for PdG. The intra-sample CV was 5.66 for $E_1C$ and 6.66 for PdG. ANCOVA tests with different dilutions from three different bonobo urine samples ($N = 3 \times 4$) confirmed that there was no violation of parallelism between the diluted samples and the standards ($P$ values ranged from 0.155 to 0.337 for $E_1C$ and 0.255 to 0.730 for PdG). The recovery test, which involved adding a fixed amount of standard solutions to three different samples ($N = 3 \times 4$), showed high regression coefficient values ($E_1C$: $Y = 0.99X + 0.01$, $r^2 = 0.985$, PdG: $Y = 0.99X - 2.5$, $r^2 = 0.967$), suggesting that $E_1C$ and PdG were successfully recovered in assays.

## Defining the fertile window and probability of ovulation and conception within a swelling cycle

The date of ovulation was estimated based on a sustained rise in urinary PdG above the baseline [59]. Briefly, the baseline was defined as the mean of urinary PdG concentration of the 10 days preceding a given day. If the PdG concentration on a given day exceeded two standard deviations above the baseline for three consecutive days within a week, the day before the first day of the sustained PdG rise was considered the ovulation date. We used a 4-day fertile window, also known as the periovulatory period [69], starting from 3 days prior to ovulation, based on findings that sperm remain

fertile within the reproductive tracts for 3 days and eggs remain fertile within 24 h after ovulation [97,98]. This criterion has been used in several studies on various primate species, including wild chimpanzees in Taï [69] and wild bonobos in Lui Kotale [50].

To calculate the probability of ovulation on a specific day from the onset of the maximal swelling phase, we used the following equation proposed in a previous study [69].

$$P(T = t) = \frac{n_t}{n}, t = 1, 2, 3\ldots$$

In this equation, $P(T = t)$ represents the probability of ovulation on a given day. Here, $t$ denotes a specific day within the maximal swelling phase, which starts from the onset of the phase. $n_t$ is the total number of cycles in which ovulation occurred on day $t$, and $n$ is the total number of ovulatory cycles, which is 14 in the current study (S2 Fig).

We also calculated the daily probability of fertility (the likelihood that the sperm cell meets the egg; fertilization) using the equation from the same study [69].

$$P(X(f) = 1) = \sum_{t=f}^{f+3} P(T = t)$$

In this equation, $P(X(f) = 1)$ represents the probability of fertility (fertilization) on a given day within the maximal swelling phase. More simply, the probability of fertilization on a given day is the sum of the probability of ovulation on the given day $f$ and the following 3 days from the given day $f$, as calculated in the previous equation for the probability of ovulation. By using this equation, we can directly compare our results between species, as well as between populations of the same species.

## Estimating male dyadic dominance hierarchy

We assessed male dominance relationships based on dyadic agonistic interactions. Agonistic interactions with physical contact included biting, hitting, and trampling, while those without physical contact included chasing, charging, and directed display (sometimes with branch dragging). We only used dyadic agonistic interactions that had a clear outcome of loser and winner. A loser was defined as the recipient of aggression who exhibited submissive behaviors such as grimacing, screaming, running away, or retreating. We excluded tied agonistic interactions from the calculation of relative dominance. In total, we observed 844 dyadic agonistic interactions. Only 16 dyadic interactions were followed by counter-aggressions. Therefore, we selected 828 dyadic interactions (SP1: 314, SP2: 162, SP3: 352) to calculate the male dominance hierarchy (S2 Table). We used the steepness package [99] to build a dominance hierarchy and calculate David's score. We also checked the h′ index [100], a modified version of Landau's linearity index [101], using the igraph package [102] to determine whether the male dominance hierarchy was linear.

## Statistical analysis

We used R 4.3.3 [103] and several packages, including lme4 [104], lmerTest [105], car [106], ggplot2 [107], and sjPlot [108] for statistical analyses and graphics. We used GLMMs or linear models (LMs) with negative binomial and binomial distributions. Mixed models that included random effects were used to control the repeated measures of behavior from the same individual. We also checked multicollinearity between independent variables in the model based on variance inflation factors (VIFs) after we ran models. In all models, the maximum VIF was less than 3, except for the nonsignificant interaction term in GLMM-1A, which had a VIF of 3.44. This indicates that multicollinearity among the predictor variables is not a serious concern [109]. All models were also significantly better in explaining the data compared with null models.

We considered that the reproductive quality of female bonobos has two main components. One is the increased chance of pregnancy as infants grow over time, irrespective of the menstrual cycle on a broad timescale. The other is changes in the probability of fertility (the chance of fertilization—a sperm meets an egg) within a menstrual cycle, depending on changes in the probability of ovulation over time. In chimpanzees, these two components are likely closely related because the energetic constraint from lactation plays a regulatory role in the resumption of the menstrual (swelling) cycle [110]. This is probably similar to female bonobos, although bonobo females resume their swelling cycle much earlier [57,59]. To integrate the effect of a dependent infant on female reproductive quality, we used infant age as a proxy for female body condition. To examine the effect of ovulation, we also included the number of days from ovulation, detumescence, or the onset of maximal swelling phase as a predictor variable in each model, depending on our question. The day of ovulation (or detumescence) was coded 0, 1 day before ovulation was −1, 2 days before was −2, and so on. The days after ovulation were 1, 2, and so on. The onset of maximal swelling phase was coded as 1, the next day was 2, and so on. In some analyses, we confined the data from −14 days before ovulation to the ovulation day (15 days, −14 to 0) to control for large inter- and intra-individual variations in the length of the maximal swelling phase. This 15-day-length maximal swelling phase was selected as the mean length of the maximal swelling phase from a published study was 13.5 ± 1.8 days (Table 3).

Infant age was coded on an ordinal scale by grouping 6 months as 0.5 years. It ranges from −1 to 5. From 6 months before parturition to the day of parturition was coded as −0.5, and the newborn infant was coded as 0. We also coded −1 from the detumescent day of maximal swelling, which coincided with ovulation, resulting in successful deliveries of an infant. Finally, to avoid high VIF and convergence errors of the model, we standardized all predictor variables to make all variables have a mean of 0 and standard deviation of 1. To make figures more intuitive and easier to interpret, they were made with the original scale.

To investigate the factors influencing a male's intensive following of a certain female, we ran two GLMMs with a negative binomial distribution, following the hurdle model approaches [111]. In the first model (GLMM-1A in Table 1), we tested whether swelling status and infant age (explanatory variables) predict the number of males (response variable) in intensive following of a specific female using all data (250 days). Female age was also included as a fixed effect to examine its influence on male intensive following. The number of males and females with maximal swelling were also included as fixed effect to examine whether the spatio-temporal distribution of males and females with maximal swelling on a given day influences the number of males who exhibit intensive following. The interaction between swelling status and infant age was also included, and female ID was the random factor in the model.

In the second part of the hurdle model, we took a subset of the data, which consisted only of those females with maximal swelling (205 days), to examine the effect of detumescence of maximal swelling and infant age (GLMM-1B in Table 1). In this model, the days from detumescence of maximal swelling were included as an explanatory variable instead of swelling scores. The other variables were the same as in GLMM-1A, except for the inclusion of the interaction between detumescent day and infant age.

We also investigated how males chose the target of the intensive following using a binomial mixed model (GLMM-1C in Table 1) with the data when there were more than 2 females with maximal swelling in a daily ranging party (77 observation days). In this model, the number of days from the onset of maximal swelling phase and infant age, as well as their interaction, were included as explanatory variables. Female age and the number of males of the day were also included, as in the two previous models.

Finally, using the data from the 14 ovulation-detected menstrual cycles (138 observation days), we tested the effect of ovulation on male intensive following of females using a negative binomial GLMM (GLMM-1D in Table 1). In this model, other explanatory variables were identical to GLMM-1B except that we changed days from detumescence to days from ovulation (ovulation day was coded 0).

To investigate the factors influencing male behaviors, including copulation, intervention in copulation, and solicitation of copulation, which are directly related to male mating efforts, we conducted four separate GLMMs with negative binomial

distribution. For these analyses, we used data from the 14 ovulation-detected menstrual cycles and limited the data from −14 to 0 days from ovulation (104 days in total). Each model (copulation: GLMM-2A and 2B, intervention: GLMM-2C, solicitation: GLMM-2D in Table 2) uses the number of copulations, interventions, and solicitations of copulation observed within a male–female dyad on a given day as a response variable. The mother–son dyad (Jk–JR) was excluded from the dataset to minimize unnecessary zero inflation in the behavioral data. The number of events was adjusted using the offset function, which included the hours that the male and female in the given dyad were observed together in the same party on a given day. The explanatory variables were the number of days from ovulation, infant age, male rank, male age, and female age. In GLMM-2B, male rank was categorized into two groups, high (1st to 3rd ranking) and low (4th to 10th ranking), to further investigate the effect of a 3-way interaction between days from ovulation, infant age, and male rank on the number of copulations. The female–male dyad nested within a certain menstrual (swelling) cycle was integrated into all models as a random variable (coded as dyad ID: cycle name).

To test whether the number of females with maximal swelling and males influences the hourly male–male aggression rate, we ran an LM (LM-2E in Table 2). As male–male aggression might be related both directly and indirectly to male mating competition, we included the presence of male intensive following (O or X) as a fixed effect, along with the number of females with maximal swelling and males on a given day. To calculate the male–male aggression rate per male per hour, we divided the total number of male–male aggressions on a given day with the total following hours of the day. We then conducted a square root transformation of the aggression rate to avoid a violation of the normality assumption of the model. We used an LM, not GLMM, as the random effect could not be defined since the number of male–male aggression incidents was pooled within a day.

## Supporting information

**S1 Fig. Properties of the target female of male's intensive following behavior (IFB).** This figure presents significant variables in GLMM-1C. The subfigures show the difference in the **(A)** infant age of females, **(B)** days from the onset of maximal swelling phase (MSP), and **(C)** age of females depending on the existence of male IFB (O) or not (X). Although GLMM-1C indicated that males followed older females more, this result might be erroneous given the distribution of female age as shown in (C). The upper and lower edges of the box represent the 75th and 25th percentiles, respectively. A line within a box is the median, and whiskers are 1.5 times the interquartile range, and colored markers are all data points of the 9 subject females. The data and R code underlying this figure can be found in the Figshare repository (https://doi.org/10.6084/m9.figshare.30403564, https://doi.org/10.6084/m9.figshare.30405073). (PDF)

**S2 Fig. 14 ovulation-detected maximal swelling phases (MSPs).** A fertile phase (periovulatory phase) is defined from −3 to 0 days from ovulation. The onset of the MSP is the first day maximal swelling was observed. For SI 1, the length (days) of MSP was defined from −18 to 0 days from ovulation. When the swelling score temporarily dropped to 2 (intermediate swelling) from 3 (maximal swelling), then rose again to 3 within 4 days, we considered MSP was continuous. The orange block represents the day that females were in the MSP (swelling 3), the white block represents not maximal swelling (swelling 1 or 2), na: absent of the female. https://doi.org/10.6084/m9.figshare.30405127. (PDF)

**S3 Fig. Effect of the 3-way interaction between ovulation, infant age, and male rank.** This figure presents the significant 3-way interaction term demonstrating that males copulated more with females with younger infants than those with older infants. Such a tendency was clearer for the three high-ranking males. Colored bands (ribbon) around each fitted line represent the 95% confidence interval (CI) from GLMM-2B. The data and R code underlying this figure can be found in the Figshare repository (https://doi.org/10.6084/m9.figshare.30403564, https://doi.org/10.6084/m9.figshare.30405190). (PDF)

**S4 Fig. Changes in hormones and swelling status of two conceptive menstrual cycles. (A)** A conceptive menstrual cycle of a female, No, that resulted in a successful delivery. An earlier surge of estrogen (E1C) than progesterone (PdG) resulted in postconception maximal swelling phase (MSP) after 3 weeks from ovulation. **(B)** A conceptive but miscarriage occurred menstrual cycle of the same female, No. Although there was a surge of estrogen and progesterone, after around 10 days from ovulation, this cycle failed in keeping a high concentration of estrogen and progesterone, so an early loss of pregnancy occurred. It is notable, however, that a short MSP came as in the successful conceptive cycle shown in (A). The data and R code underlying this figure can be found in the Figshare repository (https://doi.org/10.6084/m9.figshare.30403564, https://doi.org/10.6084/m9.figshare.30405220).
(PDF)

**S5 Fig. Changes in sexual swelling and score estimation.** Morphological changes in sexual swelling of a female, Fk. Non-swelling status was scored 1, intermediate swelling was scored 2, and the maximal swelling was scored 3. https://doi.org/10.6084/m9.figshare.30405235.
(PDF)

**S1 Table. Basic information about subject individuals in E1 group.** The age of each female in 2015 was estimated based on immigration date and several morphological cues, while the age of males was estimated mostly based on their birth date, with morphological cues. The immigration dates of Yk, Hs, Jk, and Sl were the year and month when they were identified. The infant age in April 2015 is expressed in months. Ovulation cycles: ovulation detected menstrual cycles, (anovulation): anovulatory cycle. maximal swelling phases (MSP): The number of MSPs defined for the female. †, ‡: infant births within 2 weeks. https://doi.org/10.6084/m9.figshare.30405247.
(PDF)

**S2 Table. The number of male dyadic agonistic interactions during the study periods.** In study period 1 (SP1), JR occupied the highest rank. However, the $h'$ index (0.84) indicated that the hierarchy was not linear. In SP2 & 3, the hierarchy was linear ($h' = 0.95$). The numbers next to the name represent the age of the individual in 2014 (SP1) and 2015 (SP2 & 3). https://doi.org/10.6084/m9.figshare.30405262.
(PDF)

**S1 Data. Raw data and R code underlying all statistical analyses and figures in the manuscript.** This repository contains the data and R code underlying all statistical analyses and figures presented in our study, "Male bonobo mating strategies target female fertile windows despite noisy ovulatory signals during sexual swelling." The dataset includes daily records of female swelling status and reproductive history, hormone-based estimates of ovulation timing, and male behavior measures such as intensive following, interventions, copulations, and male–male aggression. Each raw data file CSV format is named to correspond with the relevant figure number and statistical model described in the manuscript and supplementary materials. We provide analysis-ready tables, a fully reproducible R workflow for fitting all reported models, and scripts to regenerate every main and supplementary figure. To replicate the figures and statistical analyses, users can run the R code by following the instructions provided within the script. The data and R code underlying the figures and statistical analyses can be found in the Figshare repository (https://doi.org/10.6084/m9.figshare.30403564).
(ZIP)

## Acknowledgments

We extend our gratitude to our research assistants at Wamba for their invaluable assistance, and to the members of the Department of Ecology and Social Behavior and CICASP of the Primate Research Institute of Kyoto University for their support and insightful comments. We are particularly grateful to T. Sakamaki, K. Hosaka, T. Miyabe-Nishiwaki,

F. Bercovitch, N. Tokuyama, M. Nishikawa, A. MacIntosh, K. Graham, and Y. Kawaguchi for their valuable feedback on the manuscript. We also express our appreciation to the Ministry of Scientific and Technological Research of the D.R. Congo for granting research permissions, and to the Research Center of Ecology and Forestry for their assistance and for maintaining the Luo Scientific Reserve.

## Author contributions

**Conceptualization:** Heungjin Ryu, David A. Hill, Takeshi Furuichi.

**Data curation:** Heungjin Ryu.

**Formal analysis:** Heungjin Ryu.

**Funding acquisition:** Heungjin Ryu, Chie Hashimoto, Takeshi Furuichi.

**Investigation:** Heungjin Ryu, David A. Hill, Takeshi Furuichi.

**Methodology:** Heungjin Ryu, Chie Hashimoto, Keiko Mouri, Keiko Shimizu.

**Project administration:** Chie Hashimoto, Takeshi Furuichi.

**Resources:** Heungjin Ryu, Chie Hashimoto, Keiko Shimizu, Takeshi Furuichi.

**Software:** Heungjin Ryu, Keiko Mouri.

**Supervision:** David A. Hill, Takeshi Furuichi.

**Validation:** Heungjin Ryu, Keiko Mouri, Keiko Shimizu.

**Visualization:** Heungjin Ryu.

**Writing – original draft:** Heungjin Ryu.

**Writing – review & editing:** Heungjin Ryu, Chie Hashimoto, David A. Hill, Keiko Mouri, Keiko Shimizu, Takeshi Furuichi.

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
