## [Editor Report · Decision Letter 0]

17 Mar 2025

Dear Dr Ryu,

Thank you for submitting your manuscript entitled "Sexual swelling in bonobos: variations, ovulation predictability, and male responses" for consideration as a Research Article by PLOS Biology.

Your manuscript has now been evaluated by the PLOS Biology editorial staff, as well as by an academic editor with relevant expertise, and I am writing to let you know that we would like to send your submission out for external peer review.

Once your full submission is complete, your paper will undergo a series of checks in preparation for peer review. After your manuscript has passed the checks it will be sent out for review. To provide the metadata for your submission, please Login to Editorial Manager (https://www.editorialmanager.com/pbiology) within two working days, i.e. by Mar 19 2025 11:59PM.

Kind regards,

Taylor

Taylor Hart, PhD,

Associate Editor

PLOS Biology

thart@plos.org

---

## [Decision Letter · Decision Letter 1]

14 May 2025

Dear Dr Ryu,

Thank you for your patience while your manuscript "Sexual swelling in bonobos: variations, ovulation predictability, and male responses" was peer-reviewed at PLOS Biology. It has now been evaluated by the PLOS Biology editors, an Academic Editor with relevant expertise, and by several independent reviewers.

In light of the reviews, which you will find at the end of this email, we would like to invite you to revise the work to thoroughly address the reviewers' reports.

As you will see, the reviewers describe the study as well carried-out and the results as interesting. However, they all expressed concerns about the narrow focus of the study on bonobos and their differences from chimpanzees. They also all pointed out organizational and clarity issues that impeded following the results, and some additional concerns over controls, statistics, and methodological details.

Your revision will need to carefully consider and address these points. In accordance with our next point, we think that adding direct bonobo/chimpanzee comparisons is not critical here, but you should address all other concerns raised.

IMPORTANT: Crucially, for your study to be within scope for publication in PLOS Biology, you will need to substantially broaden the Introduction and Discussion sections beyond chimpanzees and bonobos. Based on our discussion with the Academic Editor, we encourage you to contextualize your study within the broader work on male mating tactics and female signals, including those done on non-primates. See also the wording from the Academic Editor at the end of this email.

Given the extent of revision needed, we cannot make a decision about publication until we have seen the revised manuscript and your response to the reviewers' comments. Your revised manuscript is likely to be sent for further evaluation by all or a subset of the reviewers.

**IMPORTANT - SUBMITTING YOUR REVISION**

*Re-submission Checklist*

*Published Peer Review*

*PLOS Data Policy*

*Blot and Gel Data Policy*

Sincerely,

Taylor

Taylor Hart, PhD,

Associate Editor

PLOS Biology

thart@plos.org

REVIEWS:

Reviewer #1: PBIOLOGY-D-25-00855: This is an interesting study examining what male know about female anogenital sexual swellings in bonobos. The study uses a strong data set to investigate the question and the methodology in general is sound. I do have some reservations detailed below howeber which need to be addressed:

Major points

1. The introduction is hugely skewed towards chimp and bonobo studies. Anogenital sexual swellings are found in a wide range of primate species and much research has been conducted into this as well as overarching theories which should be detailed in the intro. This by passes a large amount of theoretical knowledge and information from other species with anogenital swellings, I would recommend re-writing the intro to focus more on the overarching theory and wide range of species which exhibit this rather than purely chimps and bonobos.

2. Discussion is very focussed on the direct results and bonobos, there is very little comparison outside of bonobos, except to chimps every so often and broadly to primates in general at the end of the discussion. I think the paper would benefit from taking a broader perspective, highlighting other studies and integrating their findings into the discussion as with in introduction.

3. None of the hypotheses focus on chimpanzees and so the whole narrative of chimp and bonobo comparison throughout the intro seems disconnected to the aim of the study, it is not a comparative study of chimps and bonobos

4. Results: I think some basic numerical statistical output is necessary, if the results tables are all included as supplemental material then the estimate error and p value at least should be included in the text to demonstrate that a result is indeed significant, the reader should not have to hunt this info out.

5. There is a huge body of work and several analyses in the statistics, this makes the results very dense and due to the non-reporting of numerical values it can be hard to follow, huge amounts of the results have been put into supplemental materials and so the reader must flick between the results and the supplements to understand the results. This makes it a bit disjointed. On top the figures are very dense, in essence up to 9 plots in one figure which are supposed to summarise the results rather than the actual stats. I would find it easier to follow with perhaps either some whole analyses moving to the supplements or at the very least the numerical output in the text.

6. It is mentioned that mother-son relationships are present in the group, how were these controlled for in the data analysis?

7. Line 597: why was rank categorised for this analysis but not the others? It seems more reliable to use the same rank score throughout?

Minor questions:

Line 86 onwards: towards the end of the intro two of the main theories relating to the function of maximal swellings are introduced, I think these should be introduced much earlier.

Line 116: only now is it mentioned that other primates have maximal swellings.

Line 141: The figure legend here is very large, perhaps consider reducing the text, often it provides an explanation for what is in the figure as well as a basic description.

Line 167: It is unclear what the interaction term was in this analysis.

Line 232: this sentence is unclear, please rephrase

407: How many researchers collected the data? If multiple were there tests for interobserver reliability?

Line 417: Instantaneous scan sample?

Line 419: as often as possible?

609: Random effects, should a random effect related to the day of study be included here and in the other models? There could be some time related effect such as fatigue of males having to mate with multiple females across a longer time period. It might also we wise to control for party size.

Reviewer #2: This ms presents new behavioral and physiological data on a community of wild bonobos that are used to explore the ability of males to detect the ovulatory window in females, who have unusually long maximal genital swellings. The authors set up several hypotheses about behavioral variation in males in response to changes in several female attributes. The results indicate that the males are able to estimate, but not to pinpoint, the probability of ovulation. The data are hard-won and valuable, the analyses appropriate, the predictions comprehensible, and the conclusions follow from the results. Yet, it is this referee's impression that this study has produced any insights that are of interest to a large biological audience. The main comparisons made when developing the equations and aims are with chimpanzees (even though there are also detailed studies of Old World monkeys), and the long Discussion also centers mainly on bonobos. While they clearly represent an enigmatic species and are closely-related to humans, I find find such an ape-centric perspective problematic because it tends to limit the interest of this study to other great ape researchers. Thus, while the paper is technically sound, I do not see that the journal's criteria ("features works of exceptional significance, originality, and relevance; significant advances resulting from original lines of inquiry that have a broad impact in their field and across other disciplines") are satisfied here. Should the author's revise their ms for this or another journal, I have several queries and comments that may help to improve the ms (in their order of appearance):

line 58: it seems unlikely that the reduction in male competition represents an evolutionary benefit *for females* that could explain the length of their MSP (or is there a reference to this effect that can be cited?).

88: Considering the genetic similarity: something seems to be missing in this sentence

106: P1: given the small number of females/community and the long inter-birth interval, how likely is it that 2 or more females will be receptive simultaneously, ie how biologically meaningful is this prediction?

116: what is the evidence from other primates?

119: the notion of the existence of a male mating effort allocation requires the existence of costs (beyond energy and time) of matings for males. These are never discussed, questioning a key aspect of the entire analysis/paper.

418: sexual interactions were thus not only recored via focal animal sampling? Does that mean that the analyses of these key events are based on ad lib data (which would be highly problematic because of the inherent bias of that method)

Reviewer #3: This paper examines mating behavior in wild bonobos to test whether males respond to variation in female fecundity. A broader goal is to test whether bonobo females show less reliable cues of fecundity than do other primates, contributing to low rates of sexual aggression in this species.

The data and the analyses are straightforward and convincing. Males engage in more intensive following, solicitation, and competition in the period close to ovulation, and with females who have older infants (an indirect indicator of fecundity). This suggests that, like chimpanzees, bonobo males use simple behavioral rules to concentrate their mating effort on females that are likely to be more fecund. However, the data also suggest that ovulatory signals in bonobos are noisy and thus likely difficult for males to track accurately.

Although the data are clearly presented, the authors could do more to situate them in the context of previous studies. Although the discussion briefly mentions comparisons with chimpanzee data, these are not presented systematically. It would be nice to see direct and clear comparisons between these data and published chimpanzee data to be convinced that the findings here relate to species differences in aggression. Is it possible to include a table that presents such data? For example, what is the average age of a female's infant when she resumes ovulatory cycling at different sites? This appears to be younger in bonobos, meaning more cycles to conception. Table C has relevant data on differences in length of the maximal swelling phase, and this should be included in the main text, not buried in the supplemental materials. The more comparative data the authors could include to actually test the hypothesis about species differences the bettter.

The manuscript is frustrating to read, partly owing to the heavy use of unnecessary acronyms. FMS, MSP, and IFB are all used ad nauseum, forcing the reader to continually remind herself what they mean. I don't see the need for any of these acronyms, but I would pick one maximum to keep. There is no reason not to write these phrases out, or to substitute simpler phrases like "maximal swelling" or "intense following." I can't imagine many readers reading this full paper in its current version.

I don't see the need for Figure 4c, which is almost impossible to interpret, given the 3 dimensions and the opaque shading. The other figures present these data much more clearly.

The discussion could also be trimmed, as it frequently strays into speculation that is not directly supported by the data in the paper. For example, in lines 302-303 the authors state: "If their sperm is limited, they would allocate it more prudently. However, if ejaculations involved in copulation can help with refreshing sperm quality, their behavior is reasonable." This is not helpful because males might be allocating sperm prudently (i.e. investing larger ejaculates in more fecund females) - there are simply no data in this paper to test that one way or the other. If the authors want to make a case that males are mating at rates that would result in sperm depletion, they should do so with data.

----

ADVICE FROM ACADEMIC EDITOR [lightly edited]:

I agree that the work could and should be put into a larger context beyond Bonobos and Chimps, connecting to broader issues of mating tactics and sexual selection. R1 suggests connecting with other species who have anogenital swellings. I agree that discussion should be expanded, but it seems like the minimal amount of broadening. I'd encourage the authors to consider starting even broader, by connecting to the work on male mating tactics and female signals beyond primates. As is, the first paragraph of the paper briefly broadens the scope a bit, but they address the issue more narrowly in the discussion.

---

## [Editor Report · Decision Letter 2]

1 Oct 2025

Dear Dr Ryu,

Thank you for your patience while we considered your revised manuscript "Sexual swelling in bonobos: variations, ovulation predictability, and male responses" for publication as a Research Article at PLOS Biology. This revised version of your manuscript has been evaluated by the PLOS Biology editors and the Academic Editor.

Based our Academic Editor's assessment of your revision, we are likely to accept this manuscript for publication, provided you address some remaining points. The Academic Editor has provided detailed comments, which you will find below my signature -- please address them. Please also make sure to address the following data and other policy-related requests.

IMPORTANT: Please ensure that your next revision addresses the following points, as failing to do so may delay publication of your paper:

----------

**Title:

We suggest modifying the title of your paper to better emphasize the main findings and fit with our stylistic preferences. Does the following alternative formulation work for you?

"Male bonobo mating strategies target female fertile windows despite noisy ovulatory signals during sexual swelling"

**Financial disclosure statement:

Please confirm that grant numbers are included for all funding sources in you Financial Disclosure statement in the manuscript details.

**Ethics:

-- Please include the specific national or international regulations/guidelines to which your animal care and use protocol adhered. Please note that institutional or accreditation organization guidelines (such as AAALAC) do not meet this requirement.

**Data:

We have a few requests regarding your figures and data availability.

-- We see that you wrote that the data will be made available after acceptance. Please upload the data and include a link so that we can view them now.

-- Please also ensure that your uploaded data include the numerical values either in a supplementary excel file or as a permanent DOI’d deposition for the following figures:

1ABCDE

2ABCD

4AB

S1 Fig. A abc

-- Please cite the location of the data clearly in all relevant main and supplementary Figure legends, e.g. “The data underlying this Figure can be found in S1 Data” or “The data underlying this Figure can be found in https://doi.org/10.5281/zenodo.XXXXX”

-- We suggest some modifications of the naming scheme for your supplementary figures: instead of Fig. A, containing panels a, b, c, can this be changed to Fig. S1 containing panels A, B, C? This will make your paper more consistent with others at PLOS Biology and help avoid confusion among readers.

-- Supplementary files (e.g., excel). Please ensure that all data files are uploaded as 'Supporting Information' and are invariably referred to (in the manuscript, figure legends, and the Description field when uploading your files) using the following format verbatim: S1 Data, S2 Data, etc. Multiple panels of a single or even several figures can be included as multiple sheets in one excel file that is saved using exactly the following convention: S1_Data.xlsx (using an underscore).

----------

We expect to receive your revised manuscript within two weeks.

*Published Peer Review History*

*Press*

Sincerely,

Taylor

Taylor Hart, PhD,

Associate Editor

thart@plos.org

PLOS Biology

Editorial Comments from the Academic Editor:

Abstract:

Line 17-18: I suggest “This raises the question about how males successfully time mating, particularly when ovulation is difficult to predict from such signals. To address this question in bonobos, we combined…” Note that I changed “cue” to “signal”, which is the consistent with how the terms are currently used in the animal communication literature (a cue provides information incidentally, whereas a signal has been naturally selected to convey information that elicits a response from the receiver). I suggest doing so throughout the paper.

Line 20: define detumescence in a few words as part of the sentence. It’s a word most readers will need to Google. For example, “By estimating day-specific ovulation probabilities relative to the onset of the maximal swelling and subsidence of swelling (detumescence), we…”

Introduction:

I appreciate that the authors broadened the introduction further beyond bonobos. It might be a little too broad in some parts, however, as it takes a little too long to reach the key set-up for your study. I made a few suggestions for making these same points in a more succinct way.

Line 41: “…success often facilitated…” Or just drop the first sentence and begin with “Darwin’s theory of sexual selection explains how traits ..”

Lines 44-45: Some great reviews have shown that these patterns are common, but also highlight the wide diversity of sex roles. Maybe adjust phrasing to reflect this. “Males are the sex that more often develops conspicuous signals of competitive ability, and females are often choosier about their mates [1,2].” Possible citations: https://www.science.org/doi/10.1126/sciadv.1500983
https://journals.plos.org/plosbiology/article?id=10.1371/journal.pbio.3001916

Line 52-54: I don’t think the Mus example is needed.

Lines 55-57: It’s worth mentioning in this paragraph the possibility that females signal to manipulate male-male competition--either to reduce it, affect its timing, or to incite competition. In any of these cases, it’s a means of females indirectly choosing their mates (https://doi.org/10.1111/j.1558-5646.1996.tb03911.x ). Some parts of this are discussed later, but seems worth raising here, where you talk about the different reasons females may signal during reproduction.

Lines 72-73: The names of these hypotheses are confusing, though I realize that they are used commonly in the literature. The problem is that a “reliable indicator” could reliably indicate ovulation, and quality indicators are typically graded signals, so the reader needs to remember which is which. I would consider calling them the “quality indicator” (or “quality signal”) and the “ovulation signal” hypothesis. Those terms distinguish them by what the signal is indicating. You could do that by saying “The best-known hypotheses are the quality indicator hypothesis (often called reliable indicator, [27]) and the ovulation signal hypothesis (often called “graded signal”, [34]).” Or something along those lines. This is only a suggestion and up to the authors.

Line 75: It’s not clear to me why female signaling creates an arena for female-female competition. I would view the sexual signals as a consequence of the opportunity for competition. I suggest dropping the last clause, after “sexual swelling”

Line 88-90: I suggest “Despite this progress, further investigation is necessary to better understand the relative importance of mate choice and male coercion in determining fertilization, and how the conspicuity and reliability of female signals influences mating and social behaviors.”

Lines 90-92: Do you mean these are consequences of the evolution of increased conspicuity and reduced reliability? The wording implies that someone (the female or the researcher) is actively able to control the signal.

Lines 101: The paragraph above this one is a nice set up for what you plan to study here. This might therefore be a good place to shift into the “In this study…” part of the introduction. I suggest simply moving the sentence currently on line 120 (“In this study, we tested the signal reliability of the sexual swelling as a graded ovulatory signal in a wild bonobo group.”) up to the beginning of this paragraph. Replace the sentence in line 120 with something like “To examine the signal reliability of sexual swellings as a graded ovulatory signal, we first investigated…”

Line 109: Why is this called the “estrus-sex-ratio hypothesis”? I assume it increases the number of estrus females per male. This hypothesis isn’t mentioned again until the results, so the predictions aren’t made clear.

Lines 116-117: Is there evidence to support that this is a highly costly signal? I suggest “potentially costly”. The latter half of this sentence implies that sperm competition is bad for the female. Do you mean that the signal might increase male coercion? Also, the signal might hinder the female’s ability to choose her mate, but presumably not the preference itself.

---

## [Editor Report · Decision Letter 3]

31 Oct 2025

Dear Dr Ryu,

Thank you for the submission of your revised Research Article "Male bonobo mating strategies target female fertile windows despite noisy ovulatory signals during sexual swelling" for publication in PLOS Biology. On behalf of my colleagues and the Academic Editor, Gail Patricelli, I am pleased to say that we can in principle accept your manuscript for publication, provided you address any remaining formatting and reporting issues. These will be detailed in an email you should receive within 2-3 business days from our colleagues in the journal operations team; no action is required from you until then. Please note that we will not be able to formally accept your manuscript and schedule it for publication until you have completed any requested changes.

PRESS

Sincerely, 

Taylor

Taylor Hart, PhD,

Associate Editor

PLOS Biology

thart@plos.org